# Layer Flexible Adaptive Computation Time for Recurrent Neural Networks

## Abstract

Deep recurrent neural networks perform well on sequence data and are the model of choice. However, it is a daunting task to decide the structure of the networks, i.e. the number of layers, especially considering different computational needs of a sequence. We propose a layer flexible recurrent neural network with adaptive computation time, and expand it to a sequence to sequence model. Different from the adaptive computation time model, our model has a dynamic number of transmission states which vary by step and sequence. We evaluate the model on a financial data set and Wikipedia language modeling. Experimental results show the performance improvement of 7% to 12% and indicate the model's ability to dynamically change the number of layers along with the computational steps.

## 1 Introduction

Recurrent neural networks (RNN) are widely used in supervised machine learning tasks for their superior performance in sequence data, such as machine translation Auli et al. (2013); Liu et al. (2014), speech recognition Graves et al. (2013); Hannun et al. (2014), image description generation Karpathy & Fei-Fei (2015); Mao et al. (2015), and music generation Boulanger-Lewandowski et al. (2012). The design of the underlying network is always a daunting task requiring substantial computational resources and experimentation. Many recent breakthroughs hinge on multilayer neural networks ability to increase model accuracy, Hinton et al. (2012); Mohamed et al. (2012); Srivastava et al. (2015), leading to the important decision in RNNs of the number of computational steps. First, the right choice requires running several very expensive training processes to try many different computational steps. Even if a reinforcement learning algorithm is used to determine a good computational steps, Baker et al. (2017); Zoph & Le (2017), it still requires a substantial training effort. The second issue with the fixed structure RNNs is the fact that the same computational steps is applied to each input in a sequence. It is conceivable that some inputs are harder to classify than others and thus such harder inputs should employ more computational steps. A similar argument holds for steps, e.g., certain steps in a sample can bear less predictive power and thus should use fewer computational steps in order to decrease the computational burden. The goal of our work is to introduce a network that automatically determines the computational steps - and together with this the number of hidden vectors to use - in training and inference which is dynamic with respect to samples and step number.

To resolve the inherent problems of fixed structure neural networks, Graves Graves (2016) addresses this by providing an Adaptive Computation Time (ACT) model for RNN. In Graves' model, a sigmoidal halting unit is utilized to calculate a halting probability for each intermediate round within a step, and a computation stops when the accumulated halting probability reaches or exceeds a threshold. ACT can utilize multiple computation rounds within each individual step and it can dynamically adapt to different samples and steps. The model is appealing due to its modeling flexibility and its advantage in increasing model accuracy Dehghani et al. (2018). With the ACT mechanism, when a step of computation is halted, all intermediate states and outputs are used to calculate one mean-field state and output. The mean-field state and output have drawbacks. The outputs of the deepest computational step are the most informative, and should be the final outputs. The output from early computational steps may cause errors in the mean-field result which calls for using the last output only. However in such a case all computational steps should benefit from transmissions from the previous time step. This is not offered by ACT since its design is based on mean-field states and is a key feature of the proposed model. To distinguish the roles among different computational steps,

each one should obtain its computation ability and receive its state individually from the previous time step. Thus a more natural design should be a multilayer RNN with a flexible number of layers which is exactly what our proposed model offers. Our experimental results show that ACT has marginal benefits over basic RNN or sequence to sequence (seq2seq) models, indicating that ACT, with a single hidden vector, cannot always work well. This also motivates us to develop the layer flexible RNN model with adaptive computation time.

The novelty of our work is that the number of layers in our model is flexible, so that it can both achieve adaptive computation time and maintain the individual roles among different layers. Similar to Graves' work, we also utilize a unit to determine the action of each computational step within a time step by calculating their halting probabilities. To obtain the optimal computation ability, each layer should learn from the previous time step individually, and there should be concepts to decide how much to learn from each layer in the previous time step. We face the challenge that the number of layers is different between two consecutive time steps, so that we cannot set specific constant rules of how to transmit the states. In our model, each time step produces multiple hidden states (one state per computational time within the step). These multiple hidden states are then combined into a different number of hidden states for the next step using attention ideas Bahdanau et al. (2014); Luong et al. (2015) (the number of new hidden states equals to the number of computational steps in the next step). The network can thus have a flexible number of layers with dynamic number of transmission states.

In this paper, we proposed a layer flexible adaptive computation time (LFACT) model for RNNs. Each layer indicates a computational step, produces a hidden state and receives its own transmission state from the previous time step. We also extend the model to the seq2seq framework. Our experimental results show that LFACT offers significant improvements over ACT and RNN on different data sets and frameworks. With LFACT, there is no need to decide the specific structure of an RNN model through extensive experimentation, since LFACT can automatically make decisions of computational steps based on its inputs. LFACT is designed with a different logic in mind from ACT, and at the same time overcomes the problems of ACT, e.g. poor performance on certain data sets. Our model increases the accuracy of 7% to 8% on a financial data set and 12% on Wikipedia language modeling, which attests to its robustness.

The rest of the manuscript is structured as follows. In Section 2 we review the literature. In Section 3, the flexible layer adaptive computation time RNN model is presented, including all of the alternative options. In Section 4 we introduce the data sets and discuss all the experimental results.

## 2 Literature Review

A deep learning model and algorithm have many hyperparameters. In an RNN, one of the problems is deciding the computation amount of a certain input sequence. A simple solution is comparing different depths of networks and manually selecting the best option, but a series of expensive training processes is required to make the right decision. Hyperparameter optimization Bergstra et al. (2011; 2013) and Bayesian optimization Snoek et al. (2012); Mendoza et al. (2016); Saxena & Verbeek (2016) have been proposed to select an efficient architecture of a network. Based on these concepts, Zoph Zoph & Le (2017) and Baker Baker et al. (2017) propose mechanisms for network configuration using reinforcement learning. However, massive training efforts are still present. Another problem of such approaches is the assumption of a fixed structure of the network, irrespective of the underlying sample and step. The difficulty of classification varies in each data set and sample, and it is comprehensible that harder samples would require more computation. Therefore, applying networks with the same computational steps is inflexible and it cannot achieve the goal of flexible computation time among different samples. Conditional computation provides general ideas for alleviating the weaknesses of a fixed-structure deep network by establishing a learning policy Dahl et al. (2012); Bengio et al. (2015). A halt neuron is designed and used as an activation threshold in self-delimiting neural networks Schmidhuber (2012); Srivastava et al. (2013) to stop an ongoing computation whenever it reaches or exceeds the halting threshold. Work Ying & Fragkiadaki (2017) shows that conditional computation helps the networks obtain adaptive depth and thus yield higher accuracy than fixed depth structures. Graves Graves (2016) introduces an Adaptive Computation Time (ACT) mechanism for RNN to dynamically calculate each input step computation time and determine their halting condition. These series of work focus on formulating the policies of halting

conditions and use a single hidden vector in each cell; none of them contribute to designing flexible multilayer networks or study learning the rules of state transmission.

The ACT mechanism Graves (2016) is proved to improve performances and is applied in a few different problems. Universal Transformers Dehghani et al. (2018) apply ACT on a self-attentive RNN to automatically halt computation. A dynamic time model for visual attention Li et al. (2017) is proposed to accelerate the processing time by adding a binary action at each step to determine whether to continue or stop. Figurnov et al. Figurnov et al. (2017) prove that applying ACT on Residual Networks can dynamically choose the number of evaluated computational steps and propose spatially adaptive computation time for Residual Networks for image processing to adapt the computation amount between spatial positions. Similarly, Neumann et al. Neumann et al. (2016) extend ACT to a recognizing textual entailment task. In addition, ACT is also applied to reduce computation cost and calculate computation time in speech recognition Li & Liu (2018), image classification Leroux et al. (2018), natural language processing Yu et al. (2018), and highway networks Park & Yoo (2017). These models simply apply the ACT mechanism on other models to achieve the abilities of adaptive halting computations. They focus on solving their specific problems but do not make any change to the structure of ACT cells. However, our work concentrates in the inner design of a layer flexible ACT cell for its ability of automatically and dynamically adapting the number of layers.

## 3 MODEL

We start with an explanation of RNN and ACT. A standard RNN contains three layers: the input layer, the hidden layer, and the output layer. The input layer receives input sequences $x$ and transmits them to the hidden layer to compute the hidden states $u$. The output layer calculates the output $y$ based on the updated state of each step. The equations are as follows:

$$u_t = f(x_t, u_{t-1}), \qquad y_t = \sigma(W_o u_t + b_o).$$

In step $t$, input $x_t$ from the input sequence $x$ is delivered to the network. A cell in the hidden layer uses the input $x_t$ and the state $u_{t-1}$ from the previous step to update the hidden state $u_t$ in the current step. Long Short-Term Memory (LSTM) Hochreiter & Schmidhuber (1997) and Gated Recurrent Unit (GRU) Cho et al. (2014) are frequently applied in the hidden layer cell $f$, which contain the dynamic computation information and the activation of the hidden cells. The output $y_t$ is computed utilizing an output weight $W_o$, an output bias $b_o$, and an activation function $\sigma$.

ACT extends the standard RNN. The hidden layer contains several rounds of computation and each round produces an intermediate state and output. The representation of intermediate states $u_t^n$ and intermediate outputs $o_t^n$ are as follows:

$$u_t^n = \begin{cases} f(x_t^0, u_{t-1}), & n = 0 \\ f(x_t^n, u_t^{n-1}) & n > 0 \end{cases}, \qquad x_t^n = (\delta_n, x_t), \qquad o_t^n = \sigma(W_o u_t^n + b_o).$$

The first hidden cell, in step $t$, receives the state $u_{t-1}$ from the previous step $t-1$ and computes the first intermediate state. All the following rounds of computation use the previous intermediate output $u_t^{n-1}$ and produce an updated state $u_t^n$. To distinguish different rounds of computation, a flag $\delta_0$ is augmented to the input $x_t$ for the first round and another flag $\delta_n$ is added for all others. Each intermediate output $o_t^n$ is computed based on the intermediate state $u_t^n$ in the same round.

To determine the halting condition of a series of rounds of computation, units $h_t^n$ are introduced in each computation round $n$ as $h_t^n = \sigma(W_h u_t^n + b_h)$. Here $W_h$ is the halting weight and $b_h$ is the halting bias.

The total computation time $N_t$ in a step is decided by the halting units and the maximum threshold $L$. Whenever the accumulated halting units' value in a step $t$ is over 1 or the computation time reaches $L$, the computation halts. The definition of total computation time $N_t$ is as follows:

$$N_t = min\{min\{n| \sum_{i=1}^{n} h_t^i \geq 1 - \epsilon\}, L\}, \tag{1}$$

where $\epsilon$ is a hyperparameter.

ACT uses all the intermediate states and outputs to calculate one mean-field state $u_t$ and output $y_t$ (as represented in (2) and (3) below) for each step. A probability $p_t^n$ produced by halting unit $h_t^n$ is

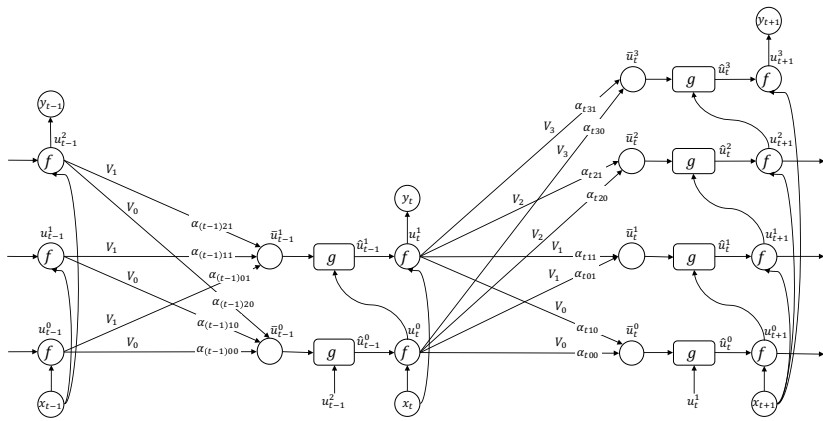

Figure 1: LFACT model - an example of three consecutive steps. Step $t-1$ has three layers, step $t$ has two layers, and step $t+1$ includes four layers.

introduced into ACT for calculating the mean-field state and output according to the contribution of each intermediate computation round in a step. The updated mean-field state $u_t$ is transmitted to the next input step and the output $o_t$ is delivered to the output layer as the current step's output.

$$p_t^n = \begin{cases} h_t^n, & n < N_t \\ 1 - \sum_{i=1}^{N_t-1} h_t^i & n = N_t \end{cases} \qquad u_t = \sum_{i=1}^{N_t} p_t^i u_t^i \quad (2) \qquad y_t = \sum_{i=1}^{N_t} p_t^i o_t^i \quad (3)$$

Given an input sequence $x$, the ACT model tends to compute as much as possible in each step to avoid making erroneous predictions and incurring errors. This can cause an extra computational expense and impede achieving the goal to adapt the computation time. Therefore, training the model to decrease the amount of computation becomes necessary. ACT introduces ponder cost $\mathcal{P}(x)$ as $\mathcal{P}(x) = N_t + p_t^{N_t}$ to represent the total computation time during the input sequence. The loss function $\mathcal{L}(x, gt)$ with $gt$ being the ground truth is modified to encourage the network to also minimize $\mathcal{P}(x)$:

$$\hat{\mathcal{L}}(x, gt) = \mathcal{L}(y(x), gt) + \tau \mathcal{P}(x) \qquad (4)$$

where $\tau$ is a hyperparameter time penalty that balances the ponder cost and prediction errors.

### 3.1 Layer Flexible Adaptive Computation Time Recurrent Neural Network

In this section, our Layer Flexible Adaptive Computation Time (LFACT) model is introduced. The main idea of LFACT is dynamically adjusting the number of layers according to the imminent characteristic of different inputs and efficiently transmitting each layer's information to the same layer in the next step. Differing from ACT where only the mean-field state $u_t$ in (2) is transmitted to the next step, which can be viewed as a single layer network, LFACT is designed for transmitting each layer's state individually between every consecutive step. In LFACT we compute $N_t$ and $N_{t+1}$ as in ACT. Each cell $n$ (layer $n$) in step $t$ takes $x_t$ and $\hat{u}_{t-1}^n$ as input and creates $u_t^n$ for $n = 1, ..., N_t$. Vector $\hat{u}_t^n$ is computed from the output $u_t^{n-1}$ of the previous cell and the hidden state $\bar{u}_{t-1}^n$ from the previous step and same layer $n$. The problem is that at step $t$ we produce $u_t^n$ for $n = 1, ..., N_t$ but for step $t+1$ we need $\bar{u}_t^n$ for $n = 1, ..., N_{t+1}$. The key of our model is to use the attention principle to create $\bar{u}_t^1, \bar{u}_t^2, ..., \bar{u}_t^{N_{t+1}}$ from $u_t^1, u_t^2, ..., u_t^{N_t}$. $Figure\ 1$ depicts the model.

The representation of the LFACT model is as follows:

$$\hat{u}_{t-1}^n = \begin{cases} g(\bar{u}_{t-1}^0, u_{t-1}^{N_{t-1}}), & n = 0 \\ g(\bar{u}_{t-1}^n, u_t^{n-1}) & n > 0 \end{cases}, \qquad u_t^n = f(x_t, \hat{u}_{t-1}^n) \ n \ge 0, \qquad o_t^n = \sigma(W_o u_t^n + b_o).$$

The LFACT model contains two types of states. One state $u_t^n$ is the primary output of each hidden cell, which is the same as the states in standard RNN. The other state is the transmission state $\bar{u}_t^n$ that is used for transmitting layer information to the next step. The primary state from previous layer

$u_t^{n-1}$ and the transmission state $\bar{u}_{t-1}^n$ from the same layer in the previous time step are combined together through function $g$. The combined state is delivered to the current cell. Possible options for $g$ are a multi-layer fully connected neural network, or an affine transformation of $(x, y)$ followed by an activation function. In our experiments, we use $g(x, y) = \sigma(W_1 x + W_2 y + b)$.

In step $t$, the hidden layer cell $f$ uses the input and the combined state from function $g$ to compute and update the primary state $u_t^n$. The primary states are used to compute the transmission state $\bar{u}_t^n$ for the next step. To avoid possible errors caused by the previous layer, input $x_t$ is directly delivered to each layer as an input. For $n \leq N_{t+1}$, the equations governing the relationship between two transmission states read

$$\bar{u}_t^n = \sum_{i=1}^{c_t^n} \alpha_{tin} u_t^n, \quad \alpha_{tin} = \frac{e^{\beta_{tin}}}{\sum_{j=1}^g e^{\beta_{tjn}}}, \quad \beta_{tin} = V_n^T \cdot \sigma(W_Q u_{t+1}^i + V_Q u_t^i + b_Q) \, i \leq c_t^n. \quad (5)$$

To compute the transmission states $\bar{u}_t^n$, an attention unit $\alpha$ is introduced to represent the relationship between the primary states $u_t^n$ in a certain layer $n$ and the primary states in other layers. We propose two choices to select $c_t^n$:

$$c_t^n = \left\{ \begin{array}{ll} min(N_t, n), & (a) \\ N_t. & (b) \end{array} \right.$$

Option (a) only considers the relationship between the state $u_t^n$ of the current layer and the states $u_t^i$ from the lower layers (i.e. $i \leq n$), called limited (LTD). Alternative (b) utilizes all computed transmission states (i.e. $i \leq N_t$), called ALL. When strategy LTD is applied and $N_{t+1} \leq N_t$, all primary states $u_t^i$ in deeper layers (i.e. $i > N_t$) cannot be used. Strategy ALL aims to include the computed information of all the layers. To distinguish different layers, extra weights $V_n$ are utilized to compute $\alpha$. Weights $V_n$, $W_Q$ and $V_Q$ in (5) to compute $\alpha$ are vectors.

We use the same method as ACT to compute $N_t$ (as represented in (1)), the computation time of each step. But unlike ACT, the halting unit is computed based on the output and transmission state of each layer as $h_t^n = \sigma(W_h u_t^n + V_h \bar{u}_{t-1}^n + b_h)$. In addition, instead of computing a mean-field output, we directly take the output of the deepest layer as one step output as $y_t = o_t^{N_t}$.

When applying loss function (4) to LFACT, the shallow layers have limited involvement in calculating gradients. Therefore, to get the prediction of each layer as accurate as possible, we introduce all of the intermediate outputs in the loss function, as

$$\widetilde{\mathcal{L}}(x, gt) = \hat{\mathcal{L}}(x, gt) + \mu \sum_{i=0}^{N_t} \bar{\mathcal{L}}(o_t^i(x), gt). \quad (6)$$

In the experiments we use $\bar{\mathcal{L}} = \mathcal{L}$.

## 3.2 Sequence to Sequence Model with LFACT

In order to deal with sequence tasks, we propose a combination model using a seq2seq (encoder-decoder) model and our LFACT model, as $Figure$ 5 in Appendix A.1 shows. In the seq2seq model, a cell in each step is replaced with our LFACT model to form a deep and flexible network. The seq2seq encoder part accepts a sequence input, and in the decoder part, we use the last ground truth as input.

## 4 Computational Experiments

All the models are trained starting with random weights, i.e. no pretraining. Training the LFACT model takes 20% to 30% more time than a typical ACT model. Most experiments are based on a single seed, but in Section 4.2 we conclude that the variance is low if the seed is varied.

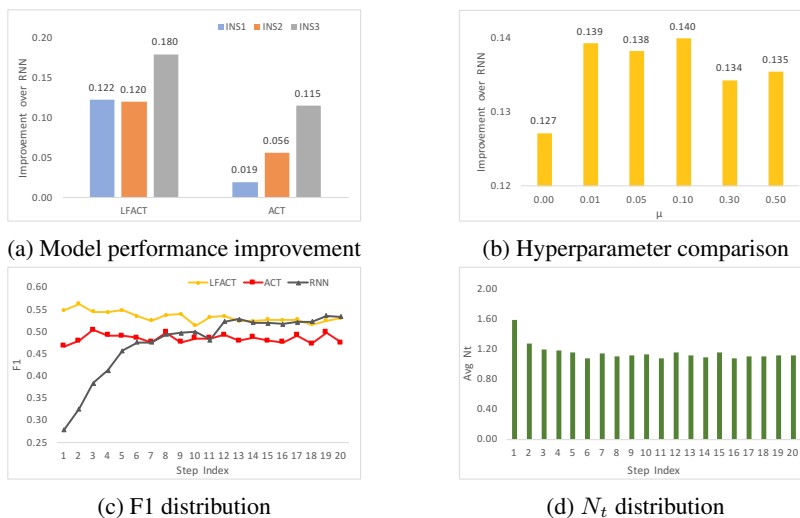

(a) Model performance improvement

(b) Hyperparameter comparison

(c) F1 distribution

(d) $N_t$ distribution

Figure 2: Results of RNN based models. (a) F1 score improvements over RNN on financial data set three instances (INS1, INS2 and INS3). (b) Average F1 improvements over RNN for different $\mu$ values on all three instances. (c) Average F1 score at each step on INS1. (d) Average computation time ($N_t$) for LFACT on INS1.

## 4.1 FINANCIAL DATA SET

We test our LFACT models on a financial data set from Harmon & Klabjan (2018). The data set consists of the tick prices of twenty-two ETFs at five minute intervals. The data is labeled into five classes to represent the significance of the price changes, e.g., one class corresponds to the price being within one standard deviation. We have 22 softmax classification layers in each step. We have three test instances, and in each one we train our model on 50 weeks of returns (45,950 samples), use the next week (905 samples) as validation data to save the best performing weights, and test the model based on the saved weights using the following week (905 samples). Sequences have lenght 20. The financial data set is tested on both RNN and seq2seq frameworks.

**RNN Based Models:** RNN based models predict the next step price changes in each time step. The LFACT model utilizes option affine transformation for $g$ ($g(x,y) = \sigma(W_1 x + W_2 y + b)$) and strategy ALL for computing transmission state $\bar{u}$ ($c_t^n = N_t$). We test plain ACT and RNN, which have been tuned with respect to all hyperparameters as our baseline models, and compare them with the RNN based LFACT model. We apply 0.001 as our ponder time penalty ($\tau = 0.001$) for LFACT and ACT (the value is obtained by the general optimal $\tau$ value of the experiments from GravesGraves (2016)), and use the Adam optimizer with 0.0005 learning rate to train the models. The maximum number of layers $L$ is 5 and GRU cells with hidden vectors of size 128 are utilized in all the models.

$Figure$ 2a shows the F1 score improvements of LFACT and ACT over RNN. We test all models on three different instances INS1, INS2, and INS3. Each bar indicates the average F1 score for all prediction steps in an instance. The results of LFACT are based on applying 0.1 to $\mu$ in loss (6). The F1 score of RNN is 0.475, 0.461, 0.447 for INS1, INS2, INS3, respectively. From $Figure$ 2a, LFACT improves 14.1% over RNN on average, and ACT improves 6.3%. We introduce the new loss function (6) in order to directly update the weights of each layer from the intermediate outputs. $Figure$ 2b provides the performance comparison for different $\mu$. The results are the average F1 score improvement over RNN for all three instances. The best range for $\mu$ in (6) is 0.01 to 0.1, and is better than the original one in (4) by 1.2%. The application of different $\mu$ values shows that our new loss function yields improvements.

$Figure$ 2c provides the F1 score distribution of steps 1 to 20 on INS1. LFACT consistently performs better than ACT, indicating that multiple layers of hidden vectors bring better effectiveness than a single one. The difficulty of a sequential prediction task is higher in early steps than in late ones, because the early steps have limited information from the input. LFACT and ACT both are stable in all prediction steps, but RNN acts poorly in early predictions. This benefit of LFACT and

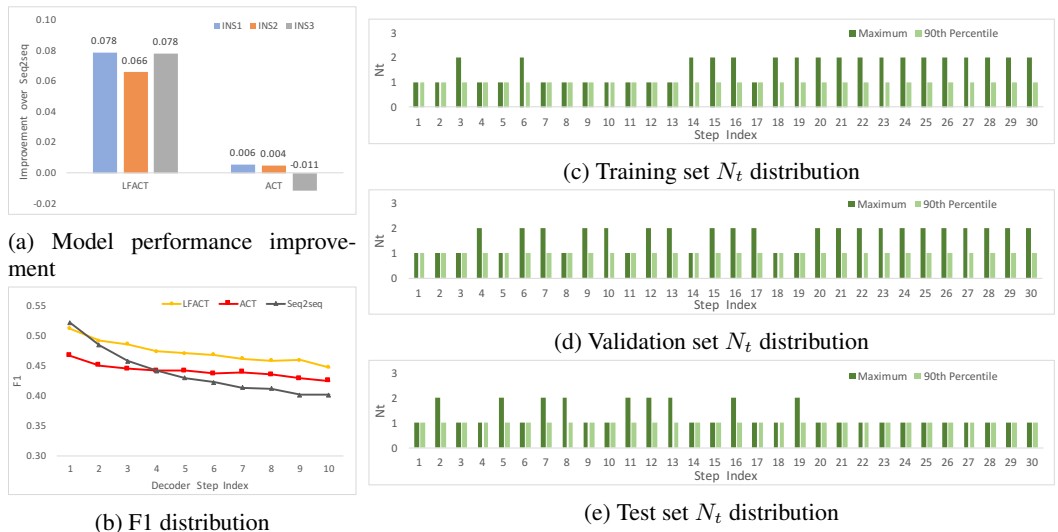

Figure 3: Results of seq2seq based models. (a) F1 score improvements over seq2seq on financial data set three instances (INS1, INS2 and INS3). (b) Average F1 score at each step on INS1. (c) (d) (e) are computation time ($N_t$) distributions based on optimized LFACT weights on INS1: X-axis is the step index; 1 to 20 indicate encoder; 21 to 30 are from the decoder part.

ACT implies that adaptive computation can contribute to hard tasks. $Figure$ 2d gives the average computation time ($N_t$) of each step on the test set of INS1. Higher average $N_t$ of early steps proves LFACT's ability of deeply computing on hard tasks, and further explains why LFACT is so effective on early predictions.

**Seq2seq Based Models (10 Prediction Steps):** In addition to the RNN framework, we also use the seq2seq version of models to predict the following ten steps. The raw sequence data with input length of 20 is delivered into seq2seq models as the inputs of the encoder part. All hyperparameters are the same as in the RNN based experiments, and the same strategies for $g$ and $c$ as in the RNN based LFACT are applied to the seq2seq framework. Considering that the encoder part does not have outputs, we apply loss function (4) in this task.

In $Figure$ 3a, we present the F1 scores relative changes over seq2seq alone for each instance. The F1 scores of seq2seq are 0.439, 0.481, 0.447. The ACT model is worse than seq2seq on INS3, so the improvement here is negative. From the results, the seq2seq based LFACT improves F1 7.4% over seq2seq, and ACT acts similar to seq2seq. In $Figure$ 3b, we provide the F1 scores for the ten prediction steps in the decoder individually on INS1. All three models decrease over time, but LFACT and ACT are more stable than seq2seq. In seq2seq based models, the decoder part has constant input of last ground truth, and can cause information deterioration as time passes. Thus, the benefits of LFACT on late predictions over seq2seq alone imply better abilities of LFACT on information transmission and memorization. Surprisingly, the first prediction of seq2seq is better than LFACT, which conflicts the results from RNN. This may be caused by LFACT requiring delay when transforming from input to predictions since it has more trainable weights than seq2seq. However, the whole point of the seq2seq framework is multiple steps of predictions, and LFACT catches up very fast at the second prediction, so the disadvantage of LFACT should not be concerning.

$Figure$ 3 also presents the computation time ($N_t$) results for INS1: $Figures$ 3c and 3d are the results of the training and validation process based on the optimized weights, and $Figure$ 3e is for test. The result shows the change of $N_t$ among the different steps, indicating that the LFACT model has the ability of adapting computation time dynamically according to its input. Because of the same input in the decoder, $N_t$ values are the same from step 21 to 30 within each set. In addition, the low $N_t$ values in test set imply that LFACT has low computation request in the decoder part. Thus, the multiple computation ability of LFACT is not the reason for the good performance in the seq2seq setting, as it is in the early predictions in the RNN setting. Comparing to seq2seq alone which contains only one computation time as well in the decoder, the significant benefits in

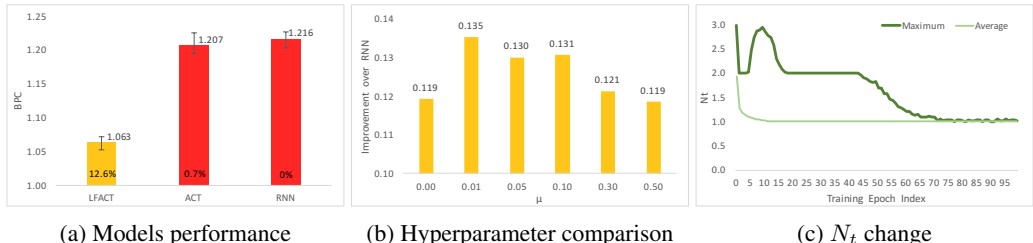

(a) Models performance  (b) Hyperparameter comparison  (c) $N_t$ change

Figure 4: Results of Wikipedia language modeling task. (a) Models performances. Numbers above bars are BPC values and the percentage inside of bars are the relative changes over RNN. (b) Average F1 improvements over RNN for different $\mu$ values. (c) Computation time ($N_t$) change during LFACT model training process.

late predictions for LFACT further confirm the conclusion that LFACT has the excellent abilities for information transmission and memorization.

We also conduct similar experiments by making 5 predictions. These are shown in Appendix A.2. The observations are very similar.

## 4.2 WIKIPEDIA LANGUAGE MODELING

This task focuses on predicting characters from the Hutter Prize Wikipedia data set, which is also used in Graves' ACT paper Graves (2016). The original unicode text is used without any preprocessing. Every character is represented as one-hot, and presents one time step. In our experiment, 10,240 sequences including 512,000 characters in total are randomly selected as the training set, and 1,280 sequences with 64,000 characters in total are chosen as validation and test sets without repetition. Each sequence includes 50 consecutive characters, and the next character is predicted at each time step in this task (RNN setting). GRU cells with 128 hidden size are used to structure all models. The maximum number of layers $L$ is set to 3, and a softmax layer with size 256 is added to each step in the decoder. We apply the optimized ponder time penalty ($\tau$) 0.06 from Graves' experiments Graves (2016) for this task. The models are evaluated using bit per character $BPC = E\left[\sum_t - \log_2 Pr(x_{t+1}|y_t)\right]$. Lower BPC values reflect better performances. All results are based on option affine transformation for g ($g(x,y) = \sigma(W_1 x + W_2 y + b)$) and strategy ALL ($c_t^n = N_t$).

In $Figure$ 4a, we present the experimental results of LFACT and the two baseline models ACT and RNN on the language modeling task. The reported BPC values for LFACT are from different settings of hyperparameter $\mu$ in loss (6). Three different random seeds are applied for ACT and RNN to test the stability of the models. Maximum, minimum, and average BPC values are provided. The bars in $Figure$ 4a represent average BPC values, and error bars indicate maximum and minimum BPC. From the experiment, ACT does not have a significant benefit over RNN, but LFACT improves 11.9% over ACT and 12.6% over standard RNN. From the error bars, LFACT has the smallest variance and ACT varies the most. Strong stability for LFACT reflects its better ability to deal with complex situations. To test the influence of the hyperparameter $\mu$ in loss function (6), we compare the different settings of $\mu$ in $Figure$ 4b. When $\mu = 0$, the loss function is equal to the original one in (4). From $Figure$ 4b, the best range for $\mu$ is from 0.01 to 0.1. However, when $\mu$ is set to be a larger value ($\mu > 0.3$), the new loss function does not bring any performance improvement over the original loss function. To assess scalability of LFACT, we also test it on different sizes of training data, as $Figure$ 9 shows in Appendix A.4. The results indicate that LFACT performs well on different sizes of training data ranging from 100,000 characters to 10 million.

In addition, we test the fully connected network option for $g(x,y)$ and strategy LTD ($c_t^n = min(N_t,n)$). The fully connected network for $g$ provides 1.074 BPC, and LTD gives 1.678. Neither of them are better than our experimental settings. Therefore, the affine transformation for $g$ and ALL are better strategies for LFACT.

In $Figure$ 4c, we provide the average maximum and average of each step computation time ($N_t$) during training of the Wikipedia language modeling task. We observe a clear decrease during the early training epochs, which eventually stabilizes. Note that during epochs 5 to 10, the maximum $N_t$ increases but the average $N_t$ still decreases. We postulate that the LFACT model has already

obtained the ability to predict most samples during this period, and is putting more effort on the difficult samples. $Figure$ 8 in Appendix A.3 shows the Maximum $N_t$ distributions of training, validation, and test based on the optimized weights. We only present the last 25 steps; the first 25 steps are all 1. The distributions show that the LFACT model is able to keep the computation time as low as possible, but also has the ability of deep computation for certain samples. With the optimized weights, only 0.03% of the sequences in the training set have more than one computation time, and validation and test sets have 0.24% and 0.16% of the sequences with multiple computation. This difference happens because the model is trained based on the training set, and the model should have learned the most efficient way to predict characters in the training set.

## 5 CONCLUSION

Deciding the structure of recurrent neural networks has been a problem in deep learning applications, in particular the number of computational steps. A halting unit is applied in a previous work to adapt the computation time to inputs, but a single hidden vector structure leads to information transmission weaknesses. We propose LFACT which utilizes an attention strategy in designing an information transmission policy which leads to a flexible multilayer recurrent neural network with adaptive computation time. LFACT can automatically adjust computation time according to the computing complexity of inputs and has outstanding dynamic information transmission abilities between consecutive time steps. We apply LFACT in an RNN and a seq2seq setting and evaluate the model on a financial data set and Wikipedia language modeling. The experimental results show a significant improvement of LFACT over RNN and seq2seq and ACT on both data sets. The different number of layers in practice indicates LFACT's ability of adapting computation time and information transmission.

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

## A    APPENDIX

### A.1    SEQ2SEQ MODEL WITH LFACT

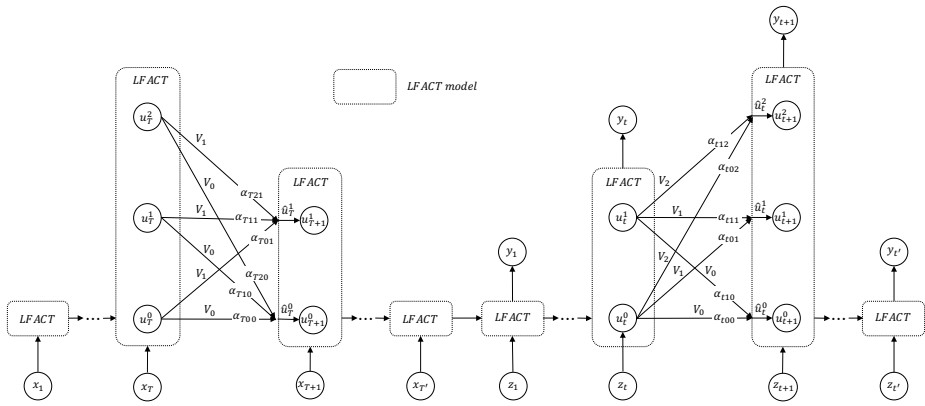

Figure 5: Seq2seq with LFACT model: the first four steps represent the encoder and the last four steps indicate the decoder. $x_1$ to $x_{T'}$ are inputs in the encoder, $z_1$ to $z_{t'}$ are inputs in the decoder, and $y_1$ to $y_{t'}$ are $t'$ steps of predictions.

### A.2    SEQ2SEQ BASED MODELS (5 PREDICTION STEPS)

To examine the stability of the LFACT model, we further test the seq2seq based models with 5 prediction steps. The setting is the same as in the 10-prediction case except we have only 5 predictions. $Figure$ 6 shows the relative F1 scores for LFACT and ACT based on seq2seq alone. The F1 scores for seq2seq on the three instances are 0.492, 0.534, and 0.498. The seq2seq based LFACT performs better than both ACT and seq2seq in the 5-prediction task, and the benefit is significant over ACT.

However, the improvement of LFACT over seq2seq is not as pronounced as in the 10-prediction task, and ACT is even worse than seq2seq. $Figure$ 7 is the F1 score distributions for the three models on INS1. The results match the 10-prediction task, and show that the advantage of LFACT is more likely to affect late predictions in the seq2seq framework.

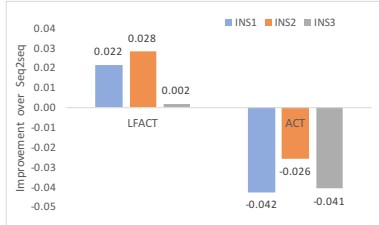

Figure 6: Performances of seq2seq based models (5 prediction steps): F1 relative changes over seq2seq on the financial data three instances (INS1, INS2, INS3).

Figure 7: F1 distributions (5 prediction steps): average F1 score at each prediction step for seq2seq based models on INS1.

### A.3 COMPUTATIONAL TIME FOR WIKIPEDIA LANGUAGE MODELING BASED ON OPTIMAL WEIGHTS

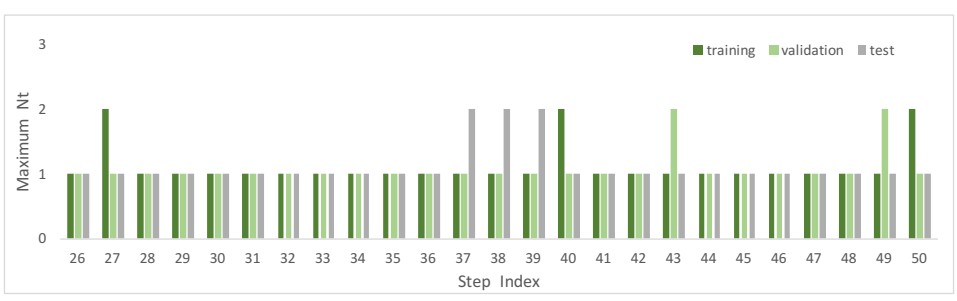

Figure 8

### A.4 PERFORMANCE ON DIFFERENT TRAINING SIZES FOR WIKIPEDIA LANGUAGE MODELING.

We test LFACT on different training sizes ranging from 100,000 characters in total to 10 million, as $Figure$ 9 shows. As the training set size increases, our model achieves better performance and eventually gets around 0.99, which indicates scalability. We conclude that LFACT consistently has over 7% improvement on all of the training sizes over ACT and RNN. Due to the computational resource limitations, all the results in $Section$ 4.2, including hyperparameter comparison, are based on 512,000 characters and 10,240 sequences training size, and 64,000 characters, 1,280 sequences test size.

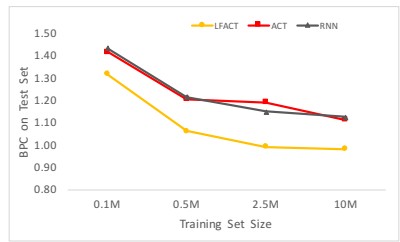

Figure 9: Performance on different training size

