# OpenReview forum: "Layer Flexible Adaptive Computation Time for Recurrent Neural Networks"
_ICLR.cc/2020/Conference — Reject_

### Official Review · AnonReviewer3 · 2019-10-23
**Official Blind Review #3**

**Rating:** 3

**Review:**

Summary:

The paper proposes to use adaptive computation time (ACT) for the number of layers (instead of the number of timesteps in the original paper) for a RNN. The intuition is that some inputs are more complex and may need more processing compare to other inputs.

Comments on the paper,

-  I am slightly confused over figure 2. For example, fig 2.c is the model performance over different steps. Does the number of steps here mean the number of time-steps per sequence? It seems that the performance of the proposed model and the RNN baseline matches at step 13. It seems to be natural to run RNNs over a sequence length of 13. it is not clear to me what is the advantage of using the proposed model in this case.

- It is also unclear to me how many layers does the RNN baseline have and how the results changes wrt the number of the layers for the baselines, i would imagine that this would get better as the number of layers increases. Can the authors compare the proposed model to RNN baselines trained with different number of layers?


- It is unclear to me what RNN-based models are, it seems that they are GRU models reading from section 4.1, but it seems to be implicit. Is there a reason to use GRUs compared to LSTMs? Do they achieve similar results?

- I am not sure the proposed datasets have been picked. Is there a reason why each dataset is picked, what would the authors expect to see (hypothesis to test) from each dataset?

- Some analytical experiments to help better understand the model would be nice. For example, some datasets would need more computational resources for some steps, but not others. Would the model be able to learn and pick that up? A really simple example if the copy task, would the model to learn to just have minimal layers for the 0's that does not contain information and would it use more layers for the steps that needs more processings (the non-zeros for example).


Minor comments:

1. some typos. P6, section 4.1, "length" was incorrectly spelled.



**Experience Assessment:**

I have published one or two papers in this area.

**Review Assessment: Checking Correctness Of Derivations And Theory:**

I assessed the sensibility of the derivations and theory.

**Review Assessment: Checking Correctness Of Experiments:**

I assessed the sensibility of the experiments.

**Review Assessment: Thoroughness In Paper Reading:**

I read the paper at least twice and used my best judgement in assessing the paper.

---

### Official Review · AnonReviewer1 · 2019-10-24
**Official Blind Review #1**

**Rating:** 3

**Review:**

Summary:

The authors propose Layer Flexible Adaptive Computation Time, an RNN-esque sequence model with varying depth at each time step. The idea is that the model can adaptively choose how much computational effort to spend on each example. The authors evaluate the model empirically on a financial dataset and Wikipedia language modeling, and find that it outperforms a vanilla RNN and the original adaptive computation time (ACT) model.

Unfortunately, the presentation of the idea is unclear, the idea itself is not very novel, and the experimental evaluation is lacking. These weaknesses lead me to vote for a weak reject.

I address specific clarity points below.

In regards to the novelty claim, there have been several developments of depth-based (as opposed to time-based) adaptive computation time in the literature, for example:

[1] McGill et al 2017 "Deciding How to Decide: Dynamic Routing in Artificial Neural Networks"
[2] Bolukbasi et al 2017 "Adaptive Neural Networks for Efficient Inference"
[3] Figurnov et al 2017 "Spatially Adaptive Computation Time for Residual Networks" (this paper is cited by the authors)

These papers do not present sequence models, but the ideas in them readily apply to sequence models. Thus, the main novelty in the authors' paper is handling the different number of hidden states at each timestep via their 'attention mechanism'. While this is definitely a contribution, the unclear presentation and lacking experimental evaluation combine to decrease the value of the paper.

Experimentally, the authors evaluate on a financial time-series dataset and Wikipedia language modeling. They compare to Adaptive computation time and a standard RNN. While the experiments demonstrate a modest improvement over ACT and an RNN, they do not compare on larger, more standard datasets such as the WMT datasets, etc... Additionally, they do not compare with other, more powerful models. Both are required to thoroughly demonstrate their model's effectiveness.

To change my mind the authors would have to (in order of importance):
1) Add experiments on WMT or other bigger datasets and compare with current SOTA models.
2) Thoroughly edit their paper for clarity (specific points below).

The authors may want also add explorations of how much computation time can be saved using their model versus others, as this is very common in the ACT-esque literature.

Specific points:
* While the original ACT paper does use 'mean-field' to denote the convex combination of states at a current timestep, that term has a specific technical meaning different from how it is used here. I would suggest using a different word.
* The introduction is too long and repeats itself in several places.
* There are repeated citations in the literature review.
* In figure 1 sometimes nodes denote functions and sometimes nodes denote outputs. Sometimes the nodes are round and sometimes they are rectangular. Sometimes arrows denote inputs to a function, and sometimes they denote multiplication. These inconsistencies make the figure very hard to decipher.
* The text description of your model is confusing. Specifically, distinguishing between the functions of u_t, \hat{u}_t and \overline{u}_t was difficult.
* In your experiments section the plots are very difficult to interpret because they are phrase as 'improvement over x'. The standard presentation is a table of absolute results. Furthermore, bar charts can be misleading because the scale can make improvements seem bigger than they actually are. The figures should stand alone without having to read the text.



**Experience Assessment:**

I have published one or two papers in this area.

**Review Assessment: Checking Correctness Of Derivations And Theory:**

I carefully checked the derivations and theory.

**Review Assessment: Checking Correctness Of Experiments:**

I assessed the sensibility of the experiments.

**Review Assessment: Thoroughness In Paper Reading:**

I read the paper thoroughly.

---

### Official Review · AnonReviewer2 · 2019-10-28
**Official Blind Review #2**

**Rating:** 3

**Review:**

This paper proposes a layer-flexible adaptive computation time model which enables learning with a different number of layers at each time step.  It proposed a set of mechanisms to make the variable layer possible. It uses attention to re-arrange the hidden states in different layers into a different number of hidden states, thus allowing the hidden layers to be variable between different time steps. It also augmented the RNN to have a transmission state that transmits layer information to the next time step.

In the LFACT model equation after Eq.4 on Page 4, the lower layer hidden states are incorporated with the previous time step hidden states through the function g, before they were sent into the RNN cell f. This makes it different than a more straight forward setting in stacked RNN where the lower layer outputs are sent to the upper layer cell directly. I am wondering why this indirect way of stacking? will it work well if the layers are stacked as the normal stacked RNNs do (i.e., u_t^n takes the lower layer hidden state u_t^{n-1} as input)?

In the experiments, it seems that the N_t is very stable through time steps. Most of them are 1 or 2 layers. Although the LFACT outperforms the RNN and ACT significantly, that could be a factor of hyperparameter tuning or model structure advantage. Have the authors tried to fix N_t as a constant (let’s say 2), and then perform the whole thing on the same setting again? I highly doubt that it will yield worse results.

For the N_t, is there any implication on what kinds of time steps should have more pondering steps and how does the model’s choice match with the expected pondering time steps? Or  at least, if we are in an unsupervised setting so that we don’t really have the “expected pondering time steps” to check the quality of adaptive layers, we should run multiple replicas of the model with different parameter initializations and show that the distribution of N_t on each of the time steps are not uniform. If the “pondering time” (N_t’s) are playing a role in processing the data, they should at least show some patterns that are related to the corresponding input, either decipherable or not.

Learning through a different number of layers between different time steps is not a novel idea. For example, the TARDIS model (https://arxiv.org/abs/1701.08718) has set the number of hidden layers variable.

In general, I think the authors have provided an interesting idea and the experiments are well performed. I’d expect the authors to isolate more factors from the model in order to show the effectiveness of the adaptive layer mechanism.


**Experience Assessment:**

I have published one or two papers in this area.

**Review Assessment: Checking Correctness Of Derivations And Theory:**

I carefully checked the derivations and theory.

**Review Assessment: Checking Correctness Of Experiments:**

I carefully checked the experiments.

**Review Assessment: Thoroughness In Paper Reading:**

I read the paper thoroughly.

---

### Decision · Program_Chairs · 2019-12-19

**Decision:**

Reject

**Comment:**

All reviewers assessed this paper as a weak reject.
The AC recommends rejection.